# Redox-Modulating Capacity and Effect of Ethyl Acetate Roots and Aerial Parts Extracts from *Geum urbanum* L. on the Phenotype Inhibition of the *Pseudomonas aeruginosa* Las/RhI Quorum Sensing System

**DOI:** 10.3390/plants14020213

**Published:** 2025-01-14

**Authors:** Lyudmila Dimitrova, Milka Mileva, Almira Georgieva, Elina Tzvetanova, Milena Popova, Vassya Bankova, Hristo Najdenski

**Affiliations:** 1The Stephan Angeloff Institute of Microbiology, Bulgarian Academy of Sciences, 1113 Sofia, Bulgaria; milkamileva@gmail.com (M.M.); almirageorgieva@gmail.com (A.G.); elina_nesta@abv.bg (E.T.); hnajdenski@abv.bg (H.N.); 2Institute of Neurobiology, Bulgarian Academy of Sciences, 1113 Sofia, Bulgaria; 3Institute of Organic Chemistry with Centre of Phytochemistry, Bulgarian Academy of Sciences, 1113 Sofia, Bulgaria; popova@orgchm.bas.bg (M.P.); bankova@orgchm.bas.bg (V.B.)

**Keywords:** *Geum urbanum* L., ethyl acetate roots and aerial pats extracts, *Pseudomonas aeruginosa*, biofilms, pyocyanin production, redox-modulating capacity, gene expression, quorum sensing

## Abstract

*Pseudomonas aeruginosa* is an opportunistic pathogen that causes nosocomial infections of the urinary tract, upper respiratory tract, gastrointestinal tract, central nervous system, etc. It is possible to develop bacteremia and sepsis in immunocompromised patients. A major problem in treatment is the development of antibiotic resistance. Therefore, new preparations of natural origin are sought, such as plant extracts, which are phytocomplexes and to which it is practically impossible to develop resistance. *Geum urbanum* L. (*Rosacea*) is a perennial herb known for many biological properties. This study aimed to investigate the redox-modulating capacity and effect of ethyl acetate (EtOAc) extracts from roots (EtOAcR) and aerial parts (EtOAcAP) of the Bulgarian plant on the phenotype inhibition of the *P. aeruginosa* Las/RhI quorum sensing (QS) system, which primarily determines drug resistance in pathogenic bacteria, including biofilm formation, motility, and pigment production. We performed QS assays to account for the effects of the two EtOAc extracts. At sub-minimal inhibitory concentrations (sub-MICs) ranging from 1.56 to 6.25 mg/mL, the biofilm formation was inhibited 85% and 84% by EtOAcR and 62% and 39% by EtOAcAP extracts, respectively. At the same sub-MICs, the pyocyanin synthesis was inhibited by 17–27% after treatment with EtOAcAP and 26–30% with EtOAcR extracts. The motility was fully inhibited at 3.12 mg/mL and 6.25 mg/mL (sub-MICs). We investigated the inhibitory potential of *las*I, *las*R, *rhi*I, and *rhi*R gene expression in biofilm and pyocyanin probes with the PCR method. Interestingly, the genes were inhibited by two extracts at 3.12 mg/mL and 6.25 mg/mL. Antiradical studies, assessed by DPPH, CUPRAC, and ABTS radical scavenging methods and superoxide anion inhibition showed that EtOAcAP extract has effective antioxidant capacity. These results could help in the development of new phytocomplexes that could be applied as biocontrol agents to inhibit the phenotype of the *P. aeruginosa* QS system and other antibiotic-resistant pathogens.

## 1. Introduction

*Pseudomonas aeruginosa* is an opportunistic human pathogen causing infections in immunocompromised patients of the respiratory system, urinary tract, gastrointestinal tract, central nervous system, eyes, ears, skin, and bones [1,2]. In 2017, carbapenem-resistant *P. aeruginosa* was the second, and in 2024, it was in the top ten as a pathogen on the WHO Bacterial Priority Pathogens List. According to the same report, the level of its resistance is high (over 30%), and the mortality rate is also high [3]. The CDC reports that more than 6000 cases of multidrug-resistant (MDR) *P. aeruginosa* occur each year out of a total of 51,000 cases in hospitals in the United States, with about 440 deaths annually. In 2017, it was reported that there were 32,600 hospitalized patients with MDR *P. aeruginosa*, of which 2700 died [4]. The proportions of carbapenem-resistant *P. aeruginosa* in the European region varied significantly. Six countries (14%) out of 44 that reported data on this microorganism in 2021 had resistance percentages equal to or higher than 50%. In 2021, Bulgaria observed an increase in the percentage of resistant strains to piperacillin-tazobactam and ceftazidime compared to previous years [5]. In some cases, this pathogen causes burn wound infection that can spread to the endogenous flora of the gastrointestinal and/or upper respiratory tract, leading to septicemia [6,7,8,9]. Cases of gangrenous cellulitis and necrotizing fasciitis developing after trauma or surgery, diabetes, and vascular insufficiency have been reported, with possible complications such as fulminant skin necrosis [10,11]. *P. aeruginosa* can adhere to damaged tissue or implanted medical devices, forming a biofilm. [12]. Biofilm formation occurs in five stages: (1) microbial cells adhesion, (2) reversible adhesion, (3) irreversible adhesion, (4) maturation, and (5) cells detachment. At the surface of the biofilm, where oxygen and nutrients are abundant, living, actively dividing bacterial cells can be found, while deeper, on the other hand, slower growing, dying cells can be found. Most antibiotics and chemotherapeutics damage actively dividing cells, because they do not destroy bacterial biofilms. Therefore, *P. aeruginosa* infections are more likely to persist and recur. Biofilm formation is associated with antimicrobial resistance (AMR), DNA transfer, carrying out metabolic activities, and other processes. These factors are regulated, and gene expression is combined in a system called quorum sensing (QS) [13]. The bacterial density coordinates the activation of the latter through the induction of specific receptor signaling complexes called autoinducers (AIs). AIs are signaling molecules synthesized during a particular growth phase or in response to specific environmental changes. When the bacterial density increases, their concentration in the environment increases accordingly. Gram-negative bacteria bind to signaling receptors located in the cytoplasm. This leads to the activation of bacterial genes for the QS system and biofilm formation [14]. *P. aeruginosa* produces three types of QS systems. Two of these are the Las (LasI synthetase and LasR transcription factor) and RhI (RhII synthetase and RhIR transcription factor) systems, which are controlled by N-acyl-homoserine lactones (AHLs). The Las system uses N-3-oxododecanoyl-homoserine lactone (3-oxo-C_12_-HSL), which is encoded by the *las*I-synthase gene, and the RhII/RhIR system uses N-butyryl-homoserine lactone (C_4_-HSL), which is encoded by the *rhl*I-synthase gene [15,16]. Between these two systems, another *Pseudomonas* quinolone system (PQS) uses 2-heptyl-3-hydroxy-4-quinolone as AI. This system is regulated by PqsR (also known as MvfR), and its activation leads to the synthesis of multiple virulence factors, such as pyocyanin and elastase [17]. PqsR positively affected the transcription of the pqsABCDE operon [18,19]. Furthermore, LasR-3-oxo-C_12_-HSL positively regulated transcription of the pqsABCDE promoter, while RhlR-C_4_-HSL negatively regulated it [19]. GacA, PQS, and Vfr positively control the Las system. It has also been demonstrated that the PQS system induces the expression of *rhl*I, which encodes the C_4_-HSL synthase. In addition, the RsaL inhibitor suppresses *las*I transcription. At a high cell density, the Las system is activated when LasR interacts with 3-oxo-C_12_-HSL, leading to the synthesis of LasB elastase, LasA protease, Aβr alkaline protease, and exotoxin A, and RhlR interacts with C_4_-HSL, leading to increased synthesis of cytotoxic lectins, pyocyanin, HCN, rhamnolipids, and swarming motility control [15,16,19] (Figure 1). The biofilm formation depends on the synthesis of 3-oxo-C_12_-HSL and C_4_-HSL [20].

Inhibition of the QS system is a process called quorum quenching (QQ) [21], a promising alternative in the fight against infections. This necessitates the search for new biologically active compounds that can disrupt the QS signaling system. Research has focused primarily on natural and synthetic agents that inhibit the action of certain AIs and gene expression. Such inhibitors are used in concentrations that do not affect bacterial growth [22].

*Geum urbanum* L., also known as St. Benedict herb, is a perennial herbaceous plant that grows in Europa, including the entire territory of Bulgaria (200–1600 m above sea level) and the Middle East [23,24]. It is also seen as a commercial product in the herbal market. Tinctures and decoctions of it have a sedative, anti-inflammatory, antiemetic, analgesic, hemostatic, and antiseptic effect, helping to heal wounds faster [25]. In Bulgarian traditional medicine, it is used to treat infections of the gastrointestinal tract (dysentery and catarrh) that are accompanied by high fevers, as well as disorders of the liver and biliary tract, intestinal colic, cough, indigestion, and vomiting. The herb is believed to be a very effective remedy for treating uterine and rectal prolapse [26]. Tinctures of inflorescences have anthelmintic and antimalarial properties [27,28]. Extracts of the species exhibit antimicrobial [29,30], anti-diabetic [31,32], and cytotoxic [33,34,35,36,37] effects due to structurally different compounds. Our previous experiments found that the ethyl acetate (EtOAc) extracts from aerial parts (EtOAcAP) and roots (EtOAcR) inhibited swarming motility and pyocyanin production of *P. aeruginosa* PA01.

Infections in living organisms caused by pathogens such as *P. aeruginosa* cause inflammation due to strong oxidative agents known as reactive oxygen species (ROS), produced mainly by phagocytic cells of the host’s immune system [38,39]. These ROS, in addition to fighting the invader in the body, damage some important cellular molecules by oxidizing (i) proteins, leading to the disruption of protein and DNA synthesis and alteration of enzyme activities; (ii) lipids, leading to membrane destabilization; and (iii) nucleic acids, causing DNA breaks [39]. There is a lot of indisputable evidence in the scientific literature that chronic infections and inflammation are associated with an increased risk of developing serious pathological changes, such as carcinogenesis. Cytokines, chemokines, prostaglandins, and reactive oxygen and nitrogen radicals circulate in the tissue microenvironment affected by chronic inflammation [40,41]. In such situations, the living organism has difficulty controlling the oxidative damage that has occurred. Antioxidant therapy comes to the rescue. As a rule, antioxidants are reducing equivalents that can act according to three main principles from their protective mechanisms: (i) preventive, (ii) radical scavenging, and (iii) restoration of enzymatic action [42]. Our previous studies have proven that, along with good antibacterial activity, the separated extracts from *G. urbanum* exhibit significant activities as antioxidants [30,34].

In the present study, we aimed to investigate the redox-modulating capacity and potential of *G. urbanum* L. EtOAcAP and EtOAcR extracts to inhibit pigment synthesis, motility, and biofilm formation of *P. aeruginosa* ATCC 27853, including *las*I, *las*R, *rhI*I, and *rhI*R gene expression. Our findings will contribute to developing new knowledge about the antibacterial mechanism of action of this Bulgarian plant, including its potential to scavenge free radicals resulting from bacterial infection in vitro.

## 2. Results

### 2.1. Antimicrobial Activity

#### 2.1.1. Determination of the Minimal Inhibitory and Bactericidal Concentrations (MIC and MBC)

The MICs and MBCs of *G. urbanum* EtOAc extracts compared to *P. aeruginosa* are presented in Table 1.

EtOAc extracts showed growth inhibitory activity between 12.5 and 25 mg/mL. For the upcoming tests, we decided to use sub-MICs in the range of 1.56–6.25 mg/mL.

#### 2.1.2. Biofilm Inhibition Potential

We investigated the biofilm formation ability of *P. aeruginosa* after treatment with EtOAc extracts without affecting bacterial growth (Figure 2).

The EtOAcR inhibited *P. aeruginosa* biofilm formation by 83.8 ± 0.03% and 84.9 ± 0.05% at concentrations of 3.12 mg/mL and 6.25 mg/mL, respectively. The same concentrations of EtOAcAP extract inhibited biofilm formation by 38.7 ± 0.04% and 61.5 ± 0.02%, respectively.

#### 2.1.3. Pyocyanin Production Quantity

Experiments were performed with sub-MICs of EtOAcAP and EtOAcR from *G. urbanum* ranging from 1.56 to 6.25 mg/mL. The results are presented in Figure 3.

The effects of the two extracts and the three tested concentrations of the extracts did not differ significantly.

#### 2.1.4. Swarming Motility Potential

The EtOAc extracts exhibited a strong inhibitory effect against the motility of *P. aeruginosa* (Figure 4).

The swarming motility of *P. aeruginosa* was completely inhibited at concentrations of 6.25 mg/mL of the tested EtOAc extracts. At concentrations of 3.12 mg/mL, the motility of *P. aeruginosa* was slightly greater after treatment with the EtOAcR extracts compared to the EtOAcAP extract.

#### 2.1.5. PCR

We performed the PCR method to demonstrate the presence of the target *rhI*I/*rhI*R and *las*I/*las*R genes in untreated bacterial culture and sub-MIC-treated (6.25 mg/mL and 3.12 mg/mL) biofilm-forming (see Section 2.1.2) and the production of the pyocyanin (see Section 2.1.3) samples (Figure 5).

The tested samples did not show the presence of the genes. It was decided not to perform real-time PCR to quantify the target genes, as PCR analysis did not reveal their presence.

### 2.2. Redox-Modulating Capacity

Table 1 contains a quantitative analysis of the total flavonoids (TFCs) and polyphenols (TPCs) in both extracts. EtOAcAP demonstrated the highest amounts of flavonoids and polyphenols compared to EtOAcR (5.92 ± 0.23 µg quercetin equivalent and 0.22 ± 0.08 and 5.92 ± 0.23 compared to 0.96 ± 0.08, respectively).

The radical-scavenging, metal-chelating, and metal-reducing properties of the extracts were expressed as IC_50_ (median of the individual effect), meaning that lower values indicate higher activity of the samples.

The ability of the extracts to reduce Fe (III) ions was determined using the FRAP method, and their activity was compared. Trolox was used as a reference substance. The EtOAcAP extract showed almost four times higher activity. A similar trend was observed for the activity associated with the reduction of copper (II)—EtOAcAP showed a twofold higher potential. Both extracts have almost the same ability to chelate Fe (II) ions. The EtOAcAP extract from *G. urbanum* has almost three times greater activity than the EtOAcR extract in scavenging DPPH radicals. Regarding superoxide anion radicals, again, EtOAcAP is the clear favorite, with an activity more than four times higher than EtOAcR. The ABTS test measures the relative ability of antioxidants to scavenge ABTS generated in the aqueous phase compared to a Trolox standard (a water-soluble vitamin E analog). In this study, it was found that the EtOAcAP extract was twice as good a scavenger of ABTS radicals as EtOAcR.

To investigate the hypoglycemic activity of both extracts, we tested their ability to inhibit the activity of the enzyme α-glucosidase. While EtOAcAP showed no activity against this enzyme, the EtOAcR extract from *G. urbanum* showed a weak ability to inhibit α-glucosidase (IC_50_ = 31.25 mg/mL) compared to the antidiabetic drug acarbose (IC_50_ = 4.2 mg/mL).

The concentration of the test sample required to inhibit 50% of the enzyme activity (IC_50_) was calculated using regression analysis.

## 3. Discussion

In the present work, we investigated the redox-modulating potential and the effect of ethyl acetate extracts from the roots and aerial parts of the Bulgarian plant *G. urbanum* L. on the phenotypic inhibition of the *P. aeruginosa* Las/RhI quorum sensing (QS) regulatory system for gene expression. We based our work on the interesting and promising results obtained previously. Therefore, the present study is a continuation and deepening of the study of the rich biological potential of the *G. urbanum* plant.

### 3.1. Antimicrobial Activity and Chemical Composition ot EtOAc Extracts

In summary, our previous studies demonstrated the potent antibacterial activity of EtOAcR and EtOAcAP extracts against *Staphylococcus aureus*, *S. epidermidis*, and *Bacillus cereus* at concentrations ranging between 39 and 2500 µg/mL [30]. The phytochemical profile of the same extracts showed a rich content of tannins, terpenoids, and flavonoids [34]. The EtOAcR extract showed a better antimicrobial effect, and we isolated several compounds (cathechin, tormentic acid, niga-ichigoside F1, gein, 3,3′-di-O-methylellagic acid-4-O-β-d-glucopyranoside, 3-O-methylellagic acid-3′-O-α-3″-O-acetylrhamnopyranoside, and 3-O-methylellagic acid-3′-O-α-2″-O-acetylrhamnopyranoside), which individually, however, did not show any activity [30]. Therefore, we performed screening tests and compared their cytotoxic potential. The EtOAcR extract was more cytotoxic than EtOAcAP, which showed a high antineoplastic activity and antiviral effect against Coxsackie B virus type 1. Both extracts were active against Human adenovirus type 5. A detailed UHPLC–HRMS (ultra-high-performance liquid chromatography-high-resolution mass spectrometry) analysis of the EtOAcAP extract was performed. It can be seen that the extract contains gallic and ellagic acid and their derivatives, methylellagic acid O-hexoside, hydroxybenzoic and hydroxycinnamic acids, acylquinic acids, phenylethanoid glycosides, and flavonoids [33]. It is known that bacterial or viral infections often trigger oncological processes in the human body [43,44,45]. There are case reports of patients with tumors and methicillin-resistant *S. aureus* (MRSA), *P. aeruginosa*, etc. [46], which are the most common cause of skin infections. Sometimes, wounds are first colonized by *S. aureus* before infection with *P. aeruginosa* [47]. We demonstrated the anti-staphylococcal activity of the EtOAcR and EtOAcAP extracts at concentrations ranging from 39 µg/mL to 1.25 mg/mL. At the same time, they have no effect against Gram-negative bacteria, such as *P. aeruginosa* [30]. Therefore, we decided to check whether the extracts contained compounds that inhibit some phenotypic manifestations of *P. aeruginosa*, such as biofilm formation, motility, and pyocyanin synthesis. A redox-active phenazine derivative, pyocyanin, is a signal for QS-mediated upregulation of virulence factors. The PQS (Figure 1) regulates the production of primary phenazine-1-carboxylic acid through the phzA-G operons. In addition to ROS production, pyocyanin functions as an electron shuttle, facilitates eDNA binding to *P. aeruginosa* cells, and influences the hydrophobicity and aggregation of cell surfaces [48]. Pyocyanin is found in clinical materials such as sputum [49,50], ear secretions [51], blood, bronchial lavage, and urine [50]. In our previous study, we demonstrated the inhibitory effect of EtOAc extracts against pyocyanin synthesis and the swarming motility of *P. aeruginosa* 01 (PA01) using Luria-Bertani medium [52]. In this experiment, we used *P. aeruginosa* ATCC 27853, grown on King’s B medium. We chose this strain because it synthesizes 2.74 µg/mL pyocyanin compared to PA01 on King’s B medium (unpublished data, 0.68 µg/mL) after 48 h of cultivation at 37 °C. King’s B media is recommended to enhance pyocyanin production [53]. Our study began with the determination of the MICs and MBCs of the extracts. The obtained concentrations (Table 1) can be applied topically (12.5–25 mg/mL), as is the case with skin infections where biofilms form. At a concentration of 6.25 mg/mL, EtOAcR extract inhibited the biofilm by 85% and EtOAcAP by 61% (Figure 2). This experiment aims to find an inhibitor of pyocyanin synthesis that does not affect bacterial growth (sub-MIC). We found that the EtOAcR and EtOAcAP extracts yielded slightly reduced pigment production (Figure 3). *P. aeruginosa* has polar flagella, which enable the bacteria to swim, swarm, and twitch. These abilities help bacteria move through aqueous environments, facilitate cell–surface interactions, and help bacteria adhere to host surfaces, leading to the formation of biofilms [54]. In this study, the swarming motility of *P. aeruginosa* was completely reduced at a sub-MIC of 6.25 mg/mL of EtOAc extracts compared to the untreated control, which showed a larger swarming diameter. At a concentration of 3.12 mg/mL, the swarming motility was reduced 3.7-fold (73% inhibition) by the EtOAcAP extract and 2.8-fold (64.6% inhibition) by the EtOAcR extract (Figure 4). Ugurlu et al. (2016) reported that phenolic acids (vanillic acid, caffeic acid, cinnamic acid, and ferulic acid) inhibited the biofilm formation (44–46%) and swarming motility (50–67%) of *P. aeruginosa* but did not affect the bacteria’s ability to produce pyocyanin (50–67%) [55]. It is known that *G. urbanum* contains vanillic acid [29,33,37,56] and caffeic acid [33,54,57]. According to our previous studies, the EtOAc extracts are rich in polyphenolic compounds [30,33]. The leaf extract of *Gynura procumbens* [58], which is similar in composition to the EtOAcAP extract from *G. urbanum* [33], inhibits biofilm and pyocyanin production in a dose-dependent manner. By using the docking method, some authors have demonstrated that quercetin, which is present in the composition of *G. urbanum* [29,54], as well as its derivatives [33,54], is a promising QQ agent [58]. Rutin [33,54] and kaempferol-3-O-rutinoside [33] also have high affinity for LasR, which is an important QS signal receptor [59]. Yang et al. (2018) reported that the dichloromethane fraction of flowers of *Camellia nitidissima* Chi [60], which is rich in gallic acid, catechin, ellagic acid, chlorogenic acid, quercetin, and kaempferol, like the *G. urbanum* EtOAcAP extract [33], inhibits pyocyanin production, swarming, and swimming motility without affecting *P. aeruginosa* PAO1 in a concentration-dependent manner (IC_50_ = 0.158 ± 0.009 mg/mL, IC_50_ = 0.139 ± 0.004 mg/mL, and IC_50_ = 0.334 ± 0.049 mg/mL, respectively). Ellagic acid and its derivatives downregulated the biofilm formation and AHL production [61]. Catechin inhibits pyocyanin production, elastase activity, biofilm formation, and the expression of the AHL synthetase genes *las*I and *rhl*I and the QS regulator genes *las*R and *rhl*R [15]. Ferulic acid also inhibits Las/RhI gene expression [62]. Interestingly, the EtOAcAP and EtOAcR extracts completely downregulated *rhI*I, *rhI*R, *las*I, and *las*R gene expression in the tested sub-MICs (Figure 5) but not the phenotypic manifestations. Therefore, we hypothesized that PQS plays a significant role in *P. aeruginosa* biofilm formation, swarming motility, and pyocyanin production.

### 3.2. Antioxidant and Redox-Modulating Activities

Intercellular bacterial communication, the so-called QS system, is a fundamental mechanism of interaction between bacterial cells in a population [63]. These facts form, so to speak, the “Gordian knot” determining the development of drug resistance in pathogenic bacteria, including biofilm formation, motility, and pigment production. According to Zhao et al. (2020), the QS communication system creates a high-density colony population and can generate small molecule signals, activate various cellular processes, tolerate antibiotics, promote drug resistance, and damage the host [64]. Often, the formation of resistant mutants is dictated by the QS system itself. Under conditions of hypoxia, the maintenance of QS communication is beneficial and even allows an increased production of toxins, including ROS [65]. Although ROS have many important functions in normal cellular metabolism, their overproduction underlies the pathological damage in many infectious diseases. Recently, inflammatory processes due to the overproduction of ROS have become one of the most important topics for the study and control of pathological conditions. The inflammatory cascade is a complex chain of molecular reactions in a living cell triggered by cell and tissue damage. It is well established that infectious diseases can provoke oxidative stress events in an infected body, as reactive oxygen and nitrogen radicals secreted by bacterial pathogens accumulate in the microenvironment of the affected tissues. In this process, the overproduction of free radicals can damage important macromolecules such as nucleic acids, proteins, and lipids. In this case, the antioxidant defense system can adequately prevent oxidative damage. Antioxidants can suppress inflammatory processes by inactivating ROS and directly blocking the activation of intracellular inflammatory signaling pathways [66]. According to Moradi et al. (2020), the use of natural and traditional plant compounds to control bacterial infections is an effective approach to interfering in their QS systems [67]. In this study, along with the antibacterial and QQ activities of the EtOAcAP and EtOAcR extracts from *G. urbanum* L., their redox-modulating properties were also evaluated in six separate systems (Table 2). We used the widely known DPPH test to assess the antioxidant activity. The origin of DPPH radicals is synthetic. The test provides valuable information regarding the ability of the studied extracts to serve as suppliers of labile-bound hydrogen in the skeletons of organic molecules [68]. The obtained results showed that EtOAcAP exhibited three times higher activity than EtOAcR. This fact is probably due to the different contents of TPC and TFC in the two extracts (Table 1). In organisms, the induction of ROS is associated with the overproduction of superoxide and hydroxyl radicals. Although superoxide anion radicals are considered to be weakly reactive radicals, they are the precursors of all other reactive species, such as H_2_O_2_, •OH, HOCl, and ONOO^−^ [69]. It is assumed that O_2_^−^• causes cell damage by participating in the iron-mediated Haber–Weiss reaction or in the superoxide-driven Fenton mechanism to generate the most reactive •OH radicals [70]. Therefore, the choice of experimental protocols included the investigation of iron (III)-reducing and iron (II)-chelating properties. The potential of EtOAcAP as a superoxide scavenger is almost four times greater than that of the root extract. Regarding the chelating effect of Fe (II), both extracts showed similar activity, with a slight superiority of EtOAcAP. Both extracts probably contain polyphenols with ortho-dihydroxyl functions substituted in their benzene rings. However, as a Fe (III) reducer, EtOAcAP showed very strong activity (IC_50_ = 4.8 µM Trolox equivalent/1 g extract). The CUPRAC method is based on the formation of a Cu(II)–neocuproine complex with antioxidants. The results of this test suggest good Cu-reducing activity of both extracts, but EtOAcAP is twice as active as EtOAcR. The mechanisms of the FRAP and CUPRAC methods are based on a single electron transfer. The reducing power of FRAP cannot detect antioxidants that act by suppressing radicals (H transfer). The FRAP method is based on the reduction of Fe (III) to Fe (II), and the CUPRAC method is based on the reduction of Cu (II) to Cu (I) [71]. When iron (III) and copper (II) are in a their “free” form, they can catalyze the production of highly toxic hydroxyl radicals by donating one electron. Therefore, the withdrawal of one electron is critical for normal cell functions [72]. In our previous study, significant amounts of good antioxidants—derivatives of gallic and ellagic acids, hydroxybenzoic and hydroxycinnamic acids (all of them with ortho-dihydroxyl substitution in the aromatic bounds), acylquinic acids, phenylethanoid glycosides, polyphenols, and flavonoids—were found in the EtOAcAP extract, which explains its better redox-modulating properties [33].

One of the most dangerous effects of ROS overproduction is ferroptosis; it is an iron-dependent oxidative cell death that various factors can cause. Ferroptosis differs from apoptosis but also results from the dysfunction of antioxidant defenses, leading to the loss of cellular redox homeostasis [73,74]. However, ferroptosis is also characterized by increased levels of intracellular ROS [75]. It has been shown that increased levels of ROS in the presence of iron ions can trigger a ferroptosis mechanism. In such a situation, high levels of ROS and ferroptosis can be effectively controlled by applying good iron chelators, e.g., deferoxamine [76].

### 3.3. Antidiabetic Effect

Alpha-glucosidases are enzymes in the digestive tract that hydrolyze carbohydrates into glucose. One strategy to combat type 2 diabetes is to inhibit the activity of α-glucosidases using natural products as an alternative option to control hyperglycemia. Serious interest has been aroused by studies to discover α-glucosidase inhibitors from plants and with the active components in them turning out to be secondary metabolites—flavonoids, polyphenols, and their glycosides [77]. Inhibition of this enzyme is one of the best strategies to reduce the postprandial rise in blood glucose and, in turn, helps to avoid the occurrence of late diabetic complications [78]. Our results showed that only EtOAcR extract exhibited activity against this enzyme. However, we cannot comment on the result at this stage. Further studies are needed to clarify. As we expected, the rich content of polyphenols, flavonoids, and tannins showed a very good antioxidant profile in the models we applied (Table 2). Our study shows that antioxidant-active extracts may be useful in reducing the risk or protecting against bacterial infection. Medicinal plants are relevant sources of antioxidants and anti-inflammatory activities for many therapeutic purposes.

## 4. Materials and Methods

### 4.1. Plant Materials and Extractions

As mentioned earlier [30], dry material from roots and aerial parts from *G. urbanum* L. were commercial products (Sunny-Yambol, Ltd.^®^, Yambol, Bulgaria). According to unpublished information, the plant was collected in April 2014 near Yambol, Bulgaria. As soon as the products were delivered, the extraction was carried out. Briefly, 500 g roots and 500 g aerial parts were macerated for 2 days in 3 L methanol and concentrated by a vacuum rotary evaporator. We extracted successively with petroleum ether, EtOAc, and n-butanol to achieve extracts of different polarity. All extracts were evaporated to dry mass. We obtained 10.4 g EtOAcAP extracts and 14.1 g EtOAcR extracts. Freshly made solutions with DMSO of the dried EtOAc extracts were used for all experiments. The toxic dose of DMSO is predetermined and taken into account, so the solvent has no effect on bacterial growth.

### 4.2. Antimicrobial Tests

#### 4.2.1. Bacterial Strains and Culture Conditions

*P. aeruginosa* ATCC (American Type Culture Collection) 27853 was cultured aerobically in brain heart infusion agar and broth (BHIA M211 and BHIB GM210, HiMedia Laboratories Pvt. Ltd., Maharashtra, India) and King’s B media (protease peptone 20 g/L, K_2_HPO_4_ 1.5 g/L, MgSO_4_.7H_2_O 1.5 g/L, agar 15 g/L or not, and glycerol 10 mL/L) overnight at 37 °C.

#### 4.2.2. Determination of the MIC and MBC

The MIC of the EtOAc extracts of the aerial parts and roots from *G. urbanum* was done by the broth microdilution method in 96-well plates with BHIB, according to ISO 20776-1:2006 [79]. Two-fold serial dilutions between 25 and 0.1 mg/mL were made for both extracts in triplicate (50 µL/well). A bacterial suspension (10^5^ CFU/mL) according to the McFarland standard was added to each well in equivalent volume. According to EUCAST guidance, gentamicin was used as a reference antibiotic. MIC is defined as the lowest drug concentration that inhibits visible bacterial growth. MBC is defined as the lowest concentration of the extract after inoculation on BHIA plates at which no bacterial growth is observed [80].

#### 4.2.3. Determination of Biofilm Inhibition Potential

The ability of *P. aeruginosa* to form biofilms at the selected sub-MICs in Section 2.1.2 (1.56–6.25 mg/mL) was tested in flat-bottomed 96-well plates, according to the protocol of Stepanović et al. [81], as we described before [82].

#### 4.2.4. Determination of Pyocyanin Production Quantity

We followed the method of Yin et al. (2015) [83] with minor modifications. Briefly, 0.5 mL of an overnight bacterial culture was inoculated into 4.5 mL of King’s B media with EtOAc extract sub-MICs (1.56–6.25 mg/mL). The probes were incubated at 37 °C for 48 h. Subsequently, a vortex was used to homogenize the samples, and they were centrifuged for 10 min at 8000 rpm and 4 °C. Three milliliters of CHC1_3_ were added to new tubes containing the supernatants. One milliliter of 0.2 M HCl (295426, Merck KGaA, Darmstadt, Germany) was added to the CHCl_3_ layer for additional extraction, and 200 µL of the resulting layer in triplicate was measured at 520 nm on 96-well plates. The following formula is used to calculate the amount of pyocyanin produced:PYO yield (µg/mL) = OD_520nm_ × 17,072, 
where OD_520nm_ is the absorption of the pigment, and PYO yield is the amount of pigment.

#### 4.2.5. Determination of Swarming Motility

Zhang et al.’s (2014) protocol [31] was applied with a few minor modifications to determine how EtOAc extracts affect *P. aeruginosa* swarming motility. Extracts at the final sub-MICs (1.56–6.25 mg/mL) were added to a pre-cooled to 55 °C 0.3% King’s B agar in a final volume of 5 mL. The suspension was quickly added to the 1.5% King’s B agar surface. The Petri dish was inoculated with a single colony using a sterile toothpick and then incubated at 37 °C overnight. The inhibitory effect of the extracts on bacterial growth was photodocumented and measured under SCAN 1200 (Interscience, Puycapel, France).

#### 4.2.6. Isolation of rRNA and Reverse Transcription PCR (RT-PCR)

The GeneMATRIX Universal RNA Purification Kit (E3598, EURx Ltd., Gdańsk, Po-land) was used to isolate total rRNA from the biofilm formation and pyocyanin production samples. Concentrations of rRNA were measured and standardized according to the manufacturer’s protocol on a Quawell UV Spectrophotometer Q3000 (Quawell Technology Inc., San Jose, CA, USA). Using random hexamer primers, reverse transcription was done with the NG dART RT kit (E0801, EURx Ltd., Gdańsk, Poland).

#### 4.2.7. PCR Analysis

The cDNA of all samples was subjected to conventional PCR. We used specific primers for each gene (Table 3) and Taq PCR Master Mix (2×) (E2520, EURx Ltd., Gdańsk, Poland). PCR reactions were denatured for 5 min at 95 °C (1 cycle) and subjected to 25 cycles at 94 °C (30 s), annealing (65 °C for 60 s), and extension at 72 °C for 1 min. A final 7-min extension at 72 °C and 4 °C to infinity was then performed. The PCR products were visualized in 1.5% agarose gels.

### 4.3. Determination of the Redox-Modulating Capacity

#### 4.3.1. Determination of the Total Flavonoids and Polyphenol Content

The total flavonoid (TFCs) and phenol contents (TPCs) was determined according to the methods of the European Pharmacopoeia [90].

#### 4.3.2. Superoxide Anion Radical Generating System (^●^O_2_^−^)

Briefly, photochemical generation of superoxide anion radicals (^●^O_2_^−^) was observed in a medium containing 50 mM potassium phosphate buffer, pH 7.8, 1.17 × 10^−6^ M riboflavin, 0.2 mM methionine, 2 × 10^−5^ M KCN, and 5.6 × 10^−5^ M nitro blue tetrazolium (NBT). The NBT reduction by superoxide to a blue formazan product, in the presence (containing different concentrations of extracts) and absence (control), was measured at 560 nm [91]. The antioxidant capacity of plant extracts was expressed as the half-maximal inhibitory concentration (IC_50_).

#### 4.3.3. DPPH Radical Scavenging Assay

The tested substances dissolved in MeOH at different concentrations reacted with the stable DPPH. After 30 min of incubation at room temperature, the reduction of DPPH● to DPPH was monitored by color changes (from deep violet to light yellow) at 517 nm [92]. The scavenging activity in percentage (AA%) was determined as follows:AA% = 100 − [(Abs_sample_ − Abs_blank_ × 100)/Abs_control_] 

#### 4.3.4. Ferric-Reducing Antioxidant Power (FRAP)

The FRAP assay was performed in the presence of TPTZ (2,4,6-Tri(2-pyridyl)-s-triazine). Basically, the method is based on the reduction of Fe (III) ion if the sample contains a reductant (antioxidant) to Fe (II) at a low pH. The colorless Fe (III)–TPTZ complex is transformed into the blue Fe (II)–TPZ complex after incubation for 4′ at 37 °C. Absorbance was measured at 593 nm [93]. The results were expressed as µmol Trolox equivalents per 1 g of extract.

#### 4.3.5. Iron-Chelating Power

The method of Lue et al. (2010) [94] was used, based on the reaction of Fe (II) ions with ferrozine, which produces a pink complex with a maximum absorption wavelength of 562 nm. The addition of 5 mM ferrozine to a sample containing a chelating agent reduced the observed absorbance. After 10 min in the dark, the absorbance was measured. The activity was defined as follows:Activity (%) = 100 (Ac − As)/(Ac), 
where Ac is the absorbance of the blank containing all compounds plus 200 µL sodium/acetate buffer. The Fe (II)-chelating ability was expressed as mM EDTA equivalent per 1 g extract.

#### 4.3.6. Cupric-Reducing Antioxidant Capacity (CUPRAC) Assay

The CUPRAC reaction is based on the forming of Cu (II)–neocuproine complex with antioxidants with an absorption maximum at 450 nm [95].

#### 4.3.7. Determination of α-Glucosidase Inhibiting Activity

The analysis was performed according to Tiwari et al. The absorption changes were measured at 405 nm against a blank solution containing p-nitrophenyl-α-d-glucopyranoside (PNG, 50 μL, 5 mM in 0.1 M phosphate buffer, pH 6.8).

Inhibition of α-glucosidase was performed according to Tewari et al. (2003) [96]. Extract solutions (100 μL) in 10% DMSO were incubated at 37 °C with Type I α-glucosidase from *Saccharomyces cerevisiae* (50 μL, 1.0 U/mL in 0.1 M phosphate buffer, pH 6.8). P-nitrophenyl-α-d-glucopyranoside (PNG, 50 μL, 5 mM in the same buffer) was added. The control, representing 100% enzyme activity, was prepared by replacing the extract with 10% DMSO, while acarbose was a positive control. Enzyme inhibition was calculated using the equationAG = (A_control_ − A_sample_)/A_control_ × 100, 
where A_control_ is the absorbance of the control mixture, and A_sample_ represents absorbance of the samples containing the extracts or acarbose.

A convenient concentration of the test sample was applied to inhibit 50% of the enzyme’s activity (IC_50_), which was calculated using regression analysis.

### 4.4. Statistical Analysis

All experiments were performed in triplicate. Experimental data were analyzed statistically using ANOVA. Data were presented as the mean ± standard deviation (SD). Statistical significance was defined as a *p*-value < 0.05.

## 5. Conclusions

Finally, we can conclude that the aerial parts and roots of *Geum urbanum* L. extracted with EtOAc inhibit 100% of the expression of the Las/RhI gene regulatory system, pyocyanin synthesis, bacterial motility, and the biofilm formation by *Pseudomonas aeruginosa* Las/RhI pathogenic factors. Therefore, we assumed that the tested phenotypic expression was significantly influenced by the third QS system, PQS. Furthermore, as free radical scavengers and metal-chelating and metal-reducing agents, both extracts showed strong redox-modulating potential. It is interesting to note that α-glucosidase was inhibited only by the EtOAcR extract. This preliminary finding warrants further investigation, because it plays a significant role in the gastrointestinal tract’s absorption of glucose in addition to understanding the mechanisms of bacterial pathogenicity, drug resistance, and adequate antioxidant protection. The current study’s findings demonstrate once again how plant extracts with a high content of secondary metabolites can have a multitargeted effect to limit the degree of damage caused by bacterial infections.

## Figures and Tables

**Figure 1 plants-14-00213-f001:**
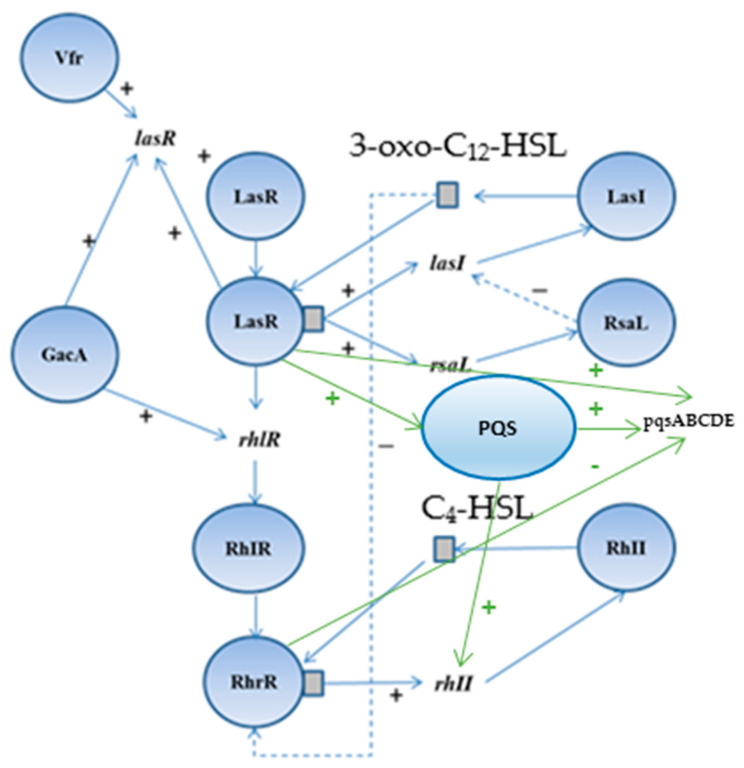
Schematic diagram showing gene regulation for the Las, Rhl, and PQS systems in *P. aeruginosa*. Legend: ‘’+’’—activation, ‘’-‘’—inhibition, Vfr—regulator of the virulence factor, GacA—global activator, LasR (*lasR* gene)—transcription activator, RhlR (*rhIR* gene)—transcription activator, LasI (*lasI* gene)—synthase of autoinducer 3-oxo-C_12_-HSL, RhII (*rhII* gene)—synthase of autoinducer C_4_-HSL, 3-oxo-C12-HSL—N-3-oxododecanoyl-homoserine lactone, C_4_-HSL—N-butyryl-homoserine lactone, PQS—*Pseudomonas* quinolone system (autoinducer is 2-heptyl-3-hydroxy-4-Quinolone), and pqsABCDE—operon that promotes PQS production.

**Figure 2 plants-14-00213-f002:**
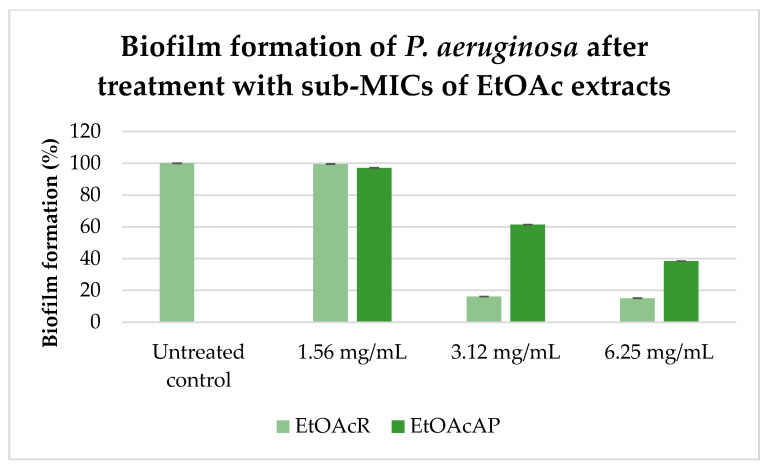
Biofilm formation of *P. aeruginosa* untreated and treated with EtOAcAP and EtOAcR from *G. urbanum*.

**Figure 3 plants-14-00213-f003:**
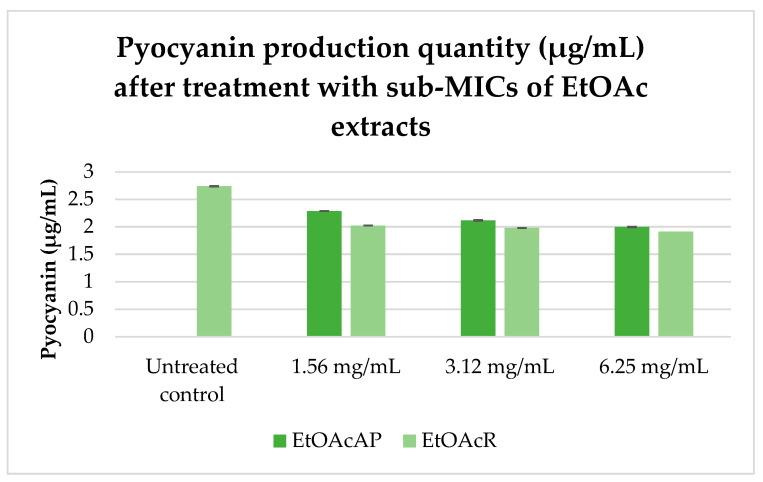
Pyocyanin production quantity of *P. aeruginosa* untreated and treated with EtOAcAP and EtOAcR from *G. urbanum*.

**Figure 4 plants-14-00213-f004:**
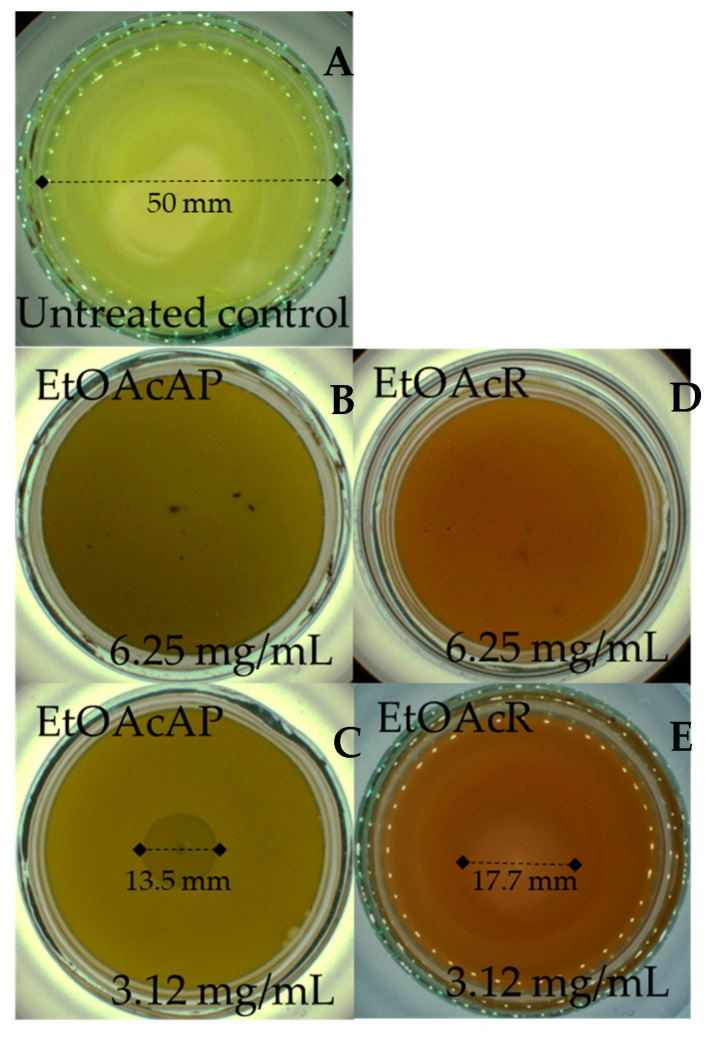
Swarming motility capacity of P. aeruginosa untreated and treated with EtOAcAP and EtOAcR from *G. urbanum*. Legend: (**A**) untreated control, (**B**) treated bacterial culture with 6.25 mg/mL of EtOAcAP extract, (**C**) treated bacterial culture with 3.12 mg/mL of EtOAcAP extract, (**D**) treated bacterial culture with 6.25 mg/mL of EtOAcR extract, and (**E**) treated bacterial culture with 3.12 mg/mL of EtOAcR extract.

**Figure 5 plants-14-00213-f005:**
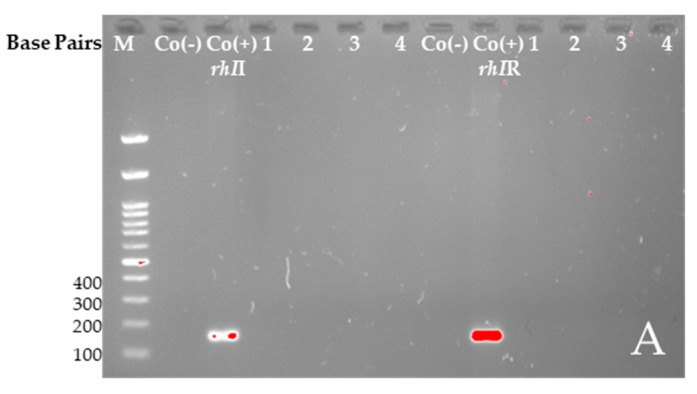
Las/RhI regulatory system gene expression by PCR from biofilm-forming and pyocyanin production probes. (**A**) *rhI*I and *rhI*R genes (M—marker; Co(−)—negative control; Co(+)—positive control. (1) EtOAcAP 6.25 mg/mL, (2) EtOAcAP 3.12 mg/mL, (3) EtOAcR 6.25 mg/mL, and (4) EtOAc 3.12 mg/mL). (**B**) *las*I and *las*R genes (M—marker; Co(−)—negative control; Co(+)—positive control. (1) EtOAcAP 6.25 mg/mL, (2) EtOAcAP 3.12 mg/mL, (3) EtOAcR 6.25 mg/mL, and (4) EtOAc 3.12 mg/mL).

**Table 1 plants-14-00213-t001:** MICs and MBCs of EtOAc extracts from roots and aerial parts of *G. urbanum*.

	MIC (mg/mL)	MBC (mg/mL)
Roots	12.5	25
Aerial parts	25	25

**Table 2 plants-14-00213-t002:** Total polyphenols, flavonoid content, radical scavenging, metal-chelating, and α-glucosidase activities of the EtOAcAP and EtOAcR extracts from *G. urbanum* L.

Extracts	TPC * [µg Galic Acid/mg Extract]	TFC ** [µg Quercetin /mg Extract]	FRAP ^#^ IC_50_	CUPRAC ^##^ IC_50_	Fe (II)-Chelating ^++^ IC_50_	DPPH Scavenging Activity, IC_50_, [mg/mL]	Superoxide Scavenging, IC_50_[mg/mL]	ABTS IC_50_[mg/mL]	Alpha-Glucosidase Activity IC_50_ [mg/mL]
Parameters									
**EtOAcAP**	5.92 ± 0.23	0.98 ± 0.05	4.8 ± 0.42	16.80± 5.87	1.50 ± 0.03	19.50 ± 1.02	4.95 ± 0.048	31.85 ± 0.26	-
**EtOAcR**	0.96 ± 0.08	0.22 ± 0.08	15.20 ± 2.13	33.3 ± 2.85	1.07 ± 0.05	67.4 ± 3.21	17.84 ± 0.15	62.25 ± 0.46	31.25 ± 0.22

* The total polyphenolic content (TPC) was calculated from the gallic acid calibration curve and expressed as µg gallic acid/mg extract. ** The total flavonoid content (TFC) was calculated from a quercetin calibration curve and expressed as µg quercetin/mg extract. ^#^ Fe (III)-reducing activity (FRAP) was calculated from a Trolox calibration curve and expressed as µM Trolox equivalent/1 g extract. ^##^ Cu (II)-reducing activity (CUPRAC) was calculated from a Trolox calibration curve and expressed as µM Trolox equivalent/1 g extract. ^++^ Fe—Fe-chelating activity was calculated from the EDTA calibration curve and expressed as mM EDTA equivalent/1 g extract. DPPH capture activity was calculated in mg/mL concentration of the extracts. α-glucosidase activity was calculated as the value of inhibition of the enzyme activity; acarbose was used as the standard.

**Table 3 plants-14-00213-t003:** List of the primers used in this study, including their sequences and melting temperatures (Tm).

Primers	Sequences	Tm	Reference
*las*I F	5′-CGT GCT CAA GTG TTC AAG G-3′	62.2 °C	[84,85]
*las*I R	5′-TAC AGT CGG AAA AGC CCA G-3′	62.8 °C	
*las*R F	5′-AAG TGG AAA ATT GGA GTG GAG-3′	61.9 °C	[84,86,87,88]
*las*R R	5′-GTA GTT GCC GAC GAC GAT GAA G-3′	67.8 °C	
*rhl*R F	5′-TGC ATT TTA TCG ATC AGG GC-3′	64.7 °C	[84,88]
*rhl*R R	5′-CAC TTC CTT TTC CAG GAC G-3′	62.3 °C	
*rhl*I F	5′-TTC ATC CTC CTT TAG TCT TCC C-3′	62.2 °C	[84,89]
*rhl*I R	5′-TTC CAG CGA TTC AGA GAG C-3′	63.3 °C	

## Data Availability

The original contributions presented in this study are included in the article. Further inquiries can be directed to the corresponding author.

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
