# Peer review of "Redox-Modulating Capacity and Effect of Ethyl Acetate Roots and Aerial Parts Extracts from *Geum urbanum* L. on the Phenotype Inhibition of the *Pseudomonas aeruginosa* Las/RhI Quorum Sensing System"

_plants, 2025, doi:10.3390/plants14020213_

Round 1

Reviewer 1 Report

Comments and Suggestions for Authors

In this manuscript, Lyudmila Dimitrova and colleagues have investigated the redox-modulating capacity and effect of ethyl acetate (EtOAc) extracts from roots (EtOAcR) and aerial parts (EtOAcAP) of the Bulgarian plant on the phenotype inhibition of the Pseudomonas aeruginosa Las/RhI quorum sensing (QS) gene expression regulatory system, which primarily determines drug resistance in pathogenic bacteria, including biofilm formation, motility, and pigment production. These results could help develop new phytocomplexes that could be applied as biocontrol agents to inhibit the phenotype of the P. aeruginosa QS system and other antibiotic-resistant pathogens. I have the following commnets:

1, For the Title, I suggest to employ “Redox-modulating capacity and effect of ethyl acetate (EtOAc) Bulgarian plant extracts on the phenotype inhibition of the Pseudomonas aeruginosa Las/RhI quorum sensing system”.

2, For the Abstract, the reason for analyzing Pseudomonas aeruginosa should be explained.

3, For the Keywords, ethyl acetate (EtOAc) extracts”, “roots” and “aerial parts” should be included.

4, For the Introduction, impact of this study should be stated in the last paragraph.

5, For the Results, significance of difference should be labelled in the Figures 2 and 3. Original gel picture for Figure 4 should be submitted.

6, For the Discussion, I would like to see that the discussion section is divided into several subsections and each subsection is entitled.

7, For the Materials and methods, genotype and growth condition of Bulgarian plants should be described in the revision.

Author Response

Dear Reviewer,

Thank you very much for your comments and suggestions! We have carefully reviewed the manuscript and completely agree with your arguments! We have tried to answer briefly and clearly. Below you will also see our responses to each of your comments:

Comments 1: For the Title, I suggest to employ “Redox-modulating capacity and effect of ethyl acetate (EtOAc) Bulgarian plant extracts on the phenotype inhibition of the Pseudomonas aeruginosa Las/RhI quorum sensing system”.

Response 1: You are right that we can change the title, but we cannot leave out the name of the plant from which the ethyl acetate extracts are derived. That is why we are changing it as follows: “Redox-modulating capacity and effect of ethyl acetate roots and aerial parts extracts from Geum urbanum L. on the phenotype inhibition of the Pseudomonas aeruginosa Las/RhI quorum sensing system”.”

Comments 2: For the Abstract, the reason for analyzing Pseudomonas aeruginosa should be explained.

Response 2: Thank you for pointing this out! We added the new text in yellow.

Comments 3: For the Keywords, “ethyl acetate (EtOAc) extracts”, “roots” and “aerial parts” should be included.

Response 3: Thank you for this comment! Since there are so many keywords, we have combined your suggestions into one (“ethyl acetate roots and aerial pats extracts”).

Comments 4: For the Introduction, the impact of this study should be stated in the last paragraph.

Response 4: Thank you for this remark! In yellow we added a sentence to the last paragraph of the Introduction.

Comments 5: For the Results, significance of difference should be labelled in the Figures 2 and 3. Original gel picture for Figure 4 should be submitted.

Response 5: Thank you very much for the recommendation! In Figure 2 and Figure 3 the statistical deviations (SD) are presented. They are very small (Figure 2, SD = 0.01 – 0.05 and Figure 3, SD = 0.004 – 0.001) and probably for this reason they are not clearly visible in the figures.

Comments 6: For the Discussion, I would like to see that the discussion section is divided into several subsections and each subsection is entitled.

Response 6: Thank you very much for pointing this out! We separate the Discussion into three subsections (3.1. Antimicrobial activity and chemical composition of EtOAc extracts, 3.2 Antioxidant and redox-modulating activities and 3.3 Antidiabetic effect).

Comments 7: For the Materials and methods, the genotype and growth condition of Bulgarian plants should be described in the revision.

Response 7: Thank you for this comment! We cannot say anything about the genotype of the plant because it is a commercial product. We added information about when the plant materials were collected.

Reviewer 2 Report

Comments and Suggestions for Authors
  • Brief summary 

Aims of the study was the investigation of the redox-modulating capacity and effect of ethyl acetate root and aerial part extracts of Geum urbanum on the phenotype inhibition of the Pseudomonas aeruginosa Las/RhI quorum sensing gene expression regulation.

  • General concept comments

    The manuscript is clear, relevant for the field and presented in a well-structured way. The study objectives are clearly stated. Experimental design is appropriate, and results are reproducible. The cited references are adequate. The conclusions are consistent with the evidence and arguments presented. Data availability statement has been reported as “Not applicable. The conclusions are consistent with the arguments presented.The manuscript addresses the areas of interest of the journal. However, I suggest a minor revision of some points.
  • Specific comments 

In the whole paper (also in references) scientific names must be in italics

Figures have a low definition and sometimes must be re-arranged

References must be accurately revised according to MDPI rules

The text should be revised to avoid spacing and typographical errors

measurement units should be reported following SI standards

Comments on the Quality of English Language

The English could be improved to more clearly express the research.

Author Response

Dear Reviewer,

Thank you very much for your positive comments and suggestions! We have carefully reviewed the manuscript and completely agree with your arguments! We have tried to answer briefly and clearly. Below you will also see our responses to each of your comments:

Comments 1: In the whole paper (also in references) scientific names must be in italics

Response 1: Thank you for pointing this out! We probably made a few mistakes when transferring the text. We have now reviewed the entire text and corrected the scientific names everywhere in Italics.

Comments 2: Figures have a low definition and sometimes must be re-arranged

Response 2: Thank you for this comment! The arrangement of the figures has been fixed! We were able to satisfactorily address your comment and corrected the figures according to the Journal Plants requirements!

Comments 3: References must be accurately revised according to MDPI rules

Response 3: Thank you very much for this comment! We used old version of EndNote. After you brought it to our attention, we manually corrected the errors.

Comments 4: The text should be revised to avoid spacing and typographical errors

Response 4: Thank you for pointing this out! We checked the whole text and fixed them!

Comments 5: Measurement units should be reported following SI standards

Response 5: Thank you very much for this suggestion! We agree with you and corrected the measurement units according to the SI standards!

Reviewer 3 Report

Comments and Suggestions for Authors

Dear Editor

Many thanks for considering me as a potential reviewer for the article " Redox-modulation capacity and phenotyping inhibitory potential of the Pseudomonas aeruginosa Las/RhI quorum sensing system by extracts of Geum urbanum L. roots and aerial parts ". The article is undoubtedly well-structured, well-presented and well-written. However, I have several observations that should be considered before proceeding further.

My observations are as follows,

·       Please change ml to mL, please do the said throughout the manuscript,

·       Line-17 Pseudomonas aeruginosa name should be italicized,

·       Lines 40-43, ‘An estimated 2,700 deaths and 32,600 hospitalized patient infections were attributed to multidrug-resistant P. aeruginosa in the United States in 2017’ what about other countries/global cases?

·       In Figure 1, please define each term/abbreviation in the figure,

·       6,25 mg/ml should typically be corrected to 6.25mg/mL, please consider this for the whole manuscript, because of international rules ‘a period (.) is used as a decimal separator, instead of (,),

·       About graphs I would like to suggest, please improve the quality and make it more refined and informative, please have a look (At figures 1 and 2); 10.3390/agriculture13020317,

·       Figure 4, you may write legends A, B C etc. and then write their details in the description of the figure,

·       4.1. Plant material and preparation of extracts A) would be nice if its changed to ‘Plant materials and extractions’  B) line165 (dry aerial and underground parts from G. urbanum) I checked Reference [29] they mention ‘roots and aerial parts’, please clearly mention which parts (leaves, stem, flowers etc.) exactly you sued for extractions? C) please write your extraction process in detail along with the quantity of material you used etc. This is very important part of your research project/paper,

·       Add a section ‘Statistical analysis’ at the end of Material and Methods,

·       Line-15 ‘effect of ethyl acetate (EtOAc) ex- 15 tracts from roots (EtOAcR) and aerial parts (EtOAcAP)’ here you refereeing extracts. Line-284 ‘aerial parts and roots of Geum urbanum L. fractionated with EtOAc’ here you are calling fractions. Look at the comparison, I think it’s not about synonyms, please try to write clear details of your extraction and its fractionations, respectively.

Comments on the Quality of English Language

Dear Editor/Authors,

I think the article is fine and well presented, however, moderate English is needed.

Thanks

Author Response

Dear Reviewer,

Thank you very much for your comments and suggestions! We have carefully reviewed the manuscript and completely agree with your arguments! We have tried to answer briefly and clearly. Below you will also see our responses to each of your comments:

Comments 1: Please change ml to mL, please do the said throughout the manuscript

Response 1: Thank you very much for this suggestion! We agree with you and corrected the measurement units according to the SI standards!

Comments 2: Line-17 Pseudomonas aeruginosa name should be italicized.

Response 2: Thank you for the comment! We probably made a few mistakes when transferring the text. We have now reviewed the entire text and corrected the scientific names everywhere to be written in Italics.

Comments 3: Lines 40-43, ‘An estimated 2,700 deaths and 32,600 hospitalized patient infections were attributed to multidrug-resistant P. aeruginosa in the United States in 2017’ what about other countries/global cases?

Response 3: We cited information from the Centers for Disease Control and Prevention webpage. This is an official website of the United States government. In yellow, we added information from the World Health Organization report, published in 2023 on the European Centre for Disease Prevention and Control webpage, about the prevalence of P. aeruginosa cases in European countries.

Comments 4: In Figure 1, please define each term/abbreviation in the figure.

Response 4: Thank toy very much for the suggestion! We added the abbreviations in a Legend.

Comments 5: 6,25 mg/ml should typically be corrected to 6.25mg/mL, please consider this for the whole manuscript, because of international rules ‘a period (.) is used as a decimal separator, instead of (,),

Response 5: Thank you very much again for this comment (See Comments 1)! We have checked the entire text and corrected the measurement units.

Comments 6: About graphs I would like to suggest, please improve the quality and make it more refined and informative, please have a look (At figures 1 and 2); 10.3390/agriculture13020317,

Response 6: Thank you for pointing this out! We were able to satisfactorily address your comment and corrected the figures according to the Journal Plants requirements!

Comments 7: Figure 4, you may write legends A, B C etc. and then write their details in the description of the figure,

Response 7: Thank you for this suggestion! The description of Figure 4 has been updated!

Comments 8: 4.1. Plant material and preparation of extracts A) would be nice if its changed to ‘Plant materials and extractions’  B) line165 (dry aerial and underground parts from G. urbanum) I checked Reference [29] they mention ‘roots and aerial parts’, please clearly mention which parts (leaves, stem, flowers etc.) exactly you sued for extractions? C) please write your extraction process in detail along with the quantity of material you used etc. This is very important part of your research project/paper.

Response 8: Thank you very much for noticing these inaccuracies! A) We renamed the Section 4.1. title to Plant materials and extractions. B) We have inadvertently made a mistake, for which we apologize. We have corrected underground parts to roots. We mention aerial parts everywhere in the text, as here stem, leaves, оflower, etc. are included. C) We rewrote the extraction process (see Section 4.1).

Comments 9: Add a section ‘Statistical analysis’ at the end of Material and Methods.

Response 9: Thank you very much for this remark! We added Section 4.4 Statistical analysis in Material and Methods.

Comments 10: Line-15 ‘effect of ethyl acetate (EtOAc) ex- 15 tracts from roots (EtOAcR) and aerial parts (EtOAcAP)’ here you refereeing extracts. Line-284 ‘aerial parts and roots of Geum urbanum L. fractionated with EtOAc’ here you are calling fractions. Look at the comparison, I think it’s not about synonyms, please try to write clear details of your extraction and its fractionations, respectively.

Response 10: Thank you very much for pointing this out! We agree that extracts and fractions are not synonyms and we have corrected the text.

Round 2

Reviewer 1 Report

Comments and Suggestions for Authors

Authors have addressed my concerns in the revision.

Author Response

Thank you very much for the positive comments! We wish you health and success in the New Year 2025!

Reviewer 3 Report

Comments and Suggestions for Authors

Dear Editor

Many thanks for the updates, regarding the article status. I am happy that the authors have considered my observations. Anyhow, below are a few minor suggestions that should be addressed,

·       Please consider my previous comment ‘Figure 4, you may write legends A, B C etc. and then write their details in the description of the figure,’ the figure is difficult to read and neither described well,

·       Line my previous comment ‘6,25 mg/ml should typically be corrected to 6.25mg/mL, please consider this for the whole manuscript, because of international rules ‘a period (.) is used as a decimal separator, instead of (,)’ please consider this for all for example, line-168 ‘83,8 ± 0,03 % and 84,9 ± 0,05 % at concentrations of’, please make the said suggestion in the whole manuscript to meet SI rules,

·       Line-12 what is ‘sowed’, please cross-check the manuscript for such minor issues,

·       Please authorize the name ‘Camellia nitidissima’,

·       Figures 2 and 3, parameters on the x-axis are not in similar orders, please maintain consistency.  

Comments on the Quality of English Language

Dear Editor/Authors,

I will suggest the authors please carefully read the article and/or consider it for minor English editing by a native English speaker!!

Thanks!!

Author Response

Dear Reviewer,

Thank you very much for your positive comments and observations! We want to wish you health and success in the New Year 2025! Below you will find our responses!  We would like to note that we have carefully checked the level of English. We believe that after the corrections we made, the text has become more understandable for readers! You can see the new changes in red in the text, as the old corrections are in yellow!

Comments 1: Please consider my previous comment ‘Figure 4, you may write legends A, B C etc. and then write their details in the description of the figure,’ the figure is difficult to read and neither described well,

Response 1: Thank you very much for this observation! We added Legend to the Figure 4! We hope this makes it easier to understand!

Comments 2: Line my previous comment ‘6,25 mg/ml should typically be corrected to 6.25mg/mL, please consider this for the whole manuscript, because of international rules ‘a period (.) is used as a decimal separator, instead of (,)’ please consider this for all for example, line-168 ‘83,8 ± 0,03 % and 84,9 ± 0,05 % at concentrations of’, please make the said suggestion in the whole manuscript to meet SI rules,

Response 2: Thank you very much for pointing this out! When making the corrections the first time, we probably missed this paragraph, for which we sincerely apologize!

Comments 3: Line-12 what is ‘sowed’, please cross-check the manuscript for such minor issues,

Response 3: Thank you for this remark! We unintentionally made a mistake when writing the manuscript. We corrected it to „showed”!

Comments 4: Please authorize the name ‘Camellia nitidissima’,

Response 4: Thank you for this comment! We moved the citation of reference № 60! We know that this plant have synonyms, such as Camellia nitidissima C.W.Chi, Camellia petelotii var. petelotii, etc., but we believe that when we citing, it is appropriate to use the name from the original publication.

Comments 5: Figures 2 and 3, parameters on the x-axis are not in similar orders, please maintain consistency.  

Response 5: Thank you for pointing this out! We have changed Figure 3 with a new one!